# TLR3 and TLR7 RNA Sensor Activation during SARS-CoV-2 Infection

**DOI:** 10.3390/microorganisms9091820

**Published:** 2021-08-26

**Authors:** Daria Bortolotti, Valentina Gentili, Sabrina Rizzo, Giovanna Schiuma, Silvia Beltrami, Giovanni Strazzabosco, Mercedes Fernandez, Francesca Caccuri, Arnaldo Caruso, Roberta Rizzo

**Affiliations:** 1Department of Chemical, Pharmaceutical and Agricultural Science, University of Ferrara, 44121 Ferrara, Italy; daria.bortolotti@unife.it (D.B.); valentina.gentili@unife.it (V.G.); sabrina.rizzo@unife.it (S.R.); giovanna.schiuma@unife.it (G.S.); silvia.beltrami@unife.it (S.B.); giovanni.strazzabosco@unife.it (G.S.); mercedes.fernandez@unife.it (M.F.); 2Department of Microbiology and Virology, “Spedali Civili,” 25126 Brescia, Italy; francesca.caccuri@unibs.it (F.C.); arnaldo.caruso@unibs.it (A.C.); 3LTTA, University of Ferrara, 44121 Ferrara, Italy

**Keywords:** SARS-CoV-2, TLR, RNA sensors

## Abstract

(1) Background: Acute respiratory syndrome coronavirus 2 (SARS-CoV-2) is the etiological agent for the coronavirus disease (COVID-19) that has led to a pandemic that began in March 2020. The role of the SARS-CoV-2 components on innate and adaptive immunity is still unknown. We investigated the possible implication of pathogen-associated molecular patterns (PAMPs)–pattern recognition receptors (PRRs) interaction. (2) Methods: We infected Calu-3/MRC-5 multicellular spheroids (MTCSs) with a SARS-CoV-2 clinical strain and evaluated the activation of RNA sensors, transcription factors, and cytokines/interferons (IFN) secretion, by quantitative real-time PCR, immunofluorescence, and ELISA. (3) Results: Our results showed that the SARS-CoV-2 infection of Calu-3/MRC-5 multicellular spheroids induced the activation of the TLR3 and TLR7 RNA sensor pathways. In particular, TLR3 might act via IRF3, producing interleukin (IL)-1α, IL-1β, IL-4, IL-6, and IFN-α and IFN-β, during the first 24 h post-infection. Then, TLR3 activates the NFκB transduction pathway, leading to pro-inflammatory cytokine secretion. Conversely, TLR7 seems to mainly act via NFκB, inducing type 1 IFN, IFN-γ, and IFN-λ3, starting from the 48 h post-infection. (4) Conclusion: We showed that both TLR3 and TLR7 are involved in the control of innate immunity during lung SARS-CoV-2 infection. The activation of TLRs induced pro-inflammatory cytokines, such as IL-1α, IL-1β, IL-4, and IL-6, as well as interferons. TLRs could be a potential target in controlling the infection in the early stages of the disease.

## 1. Introduction

SARS-CoV-2 is a new strain of the positive ssRNA coronavirus family, the cause of the coronavirus disease (COVID-19), which shares high homology with the previous severe disease-associated coronaviruses MERS (Middle East respiratory syndrome) and SARS (severe acute respiratory syndrome). The SARS-CoV-2 outbreak emerged firstly in Wuhan in December 2019, and then rapidly spread worldwide, becoming a pandemic in January 2020 [1,2,3].

As reported by different published works, COVID-19 is associated with a peculiar clinical case history, characterized by an inefficient immune system response and high levels of inflammatory cytokines, known as “cytokine storm”, including IL-1, IL-6, IL-4, IL-10, and INF-γ [4]. In particular, the presence of high serum levels of these cytokines has been associated with severe COVID-19 [5], reported to be possibly associated with the increased expression of angiotensin converting enzyme 2 (ACE2), which is the cellular receptor bound by SARS-CoV-2 spike (S) protein. The ACE2 receptor is essential for viral entry into the target cells [6]. Simultaneously, the interaction between ACE2 and SARS-CoV-2 increases the secretion of soluble ACE2 in the blood and urine [7], leading to the release of a massive quantity of cytokines (cytokine storm), which includes the production of IL-6 by macrophages [8,9]. This condition is related to severe lymphocytopenia [10], hypercoagulation [11], increased mortality [12], and a poor clinical follow-up [13]. The typical inflammatory environment triggered by SARS-CoV-2 infection is the result of the initial recruitment of the innate immune response, which represents the first line of protection against pathogens and that, in turn, stimulates acquired immunity activation [14].

For this reason, to achieve efficient control of the infection, it is crucial that the host immune response is balanced, in order to avoid both excessive inflammation that could damage the host system, as observed in COVID-19 patients’ lungs, and low activation of the immune system, which could facilitate viral spread [15,16].

During a viral infection, both the infected cells and the innate immune system are aware of the presence of an infection, by the recognition of specific pathogen portions, called “pathogen-associated molecular patterns” (PAMPs), which are recognized by specific pattern recognition receptors (PRRs) [17]. The PAMPs–PRRs interaction leads to an intracellular signaling cascade that is essential for both the antiviral activity by interferons production, and immune system activation by cytokine secretion [17]. The PRRs family includes different components that are involved in the sensing of RNA virus infections, such as RIG-I-like receptors (RLRs), e.g., RIG-I and MDA5, and Toll-like receptors (TLRs) [18]. RLRs consist of cytoplasmatic RNA helicases that recognize intracellular double-stranded RNA (dsRNA), while TLRs are membrane-associated receptors that are able to recognize PAMPs [19,20]. Anti-viral TLRs include TLR3, which recognizes dsRNA, TLR7 and TLR8 that engage single-stranded RNA (ssRNA), while TLR9 detects unmethylated CpG DNA. In particular, TLRs 3, 7, and 8 are all localized on the endosomal membrane, and could recognize ssRNA [21]. TLR3 engagement by viral dsRNA activates the TRIF-dependent pathway and induces proinflammatory cytokines, chemokines, and type I and type III interferons via NF-κB and IRF3. TLR7/8, located on the X-chromosome, as tandem duplicated genes, are expressed on the endosome membranes. The interaction between TLR7/8 and ssRNA enhances immune activation and the release of pro-inflammatory molecules, which might be connected with disease outcome [21]. Recently, TLR7 has been reported to be implicated in the sensing of SARS-CoV-2 infection, and the presence of TLR7-deficient genetic variants have been associated with a less-efficient control of the infection [22]. This central role of TLR7 in the antiviral response towards SARS-CoV-2 has been considered to be a potential target for therapy with the immune-stimulator imiquimod, in order to increase TLR7 activation and, consequently, its antiviral effect [22]. Once the viral RNA sensors are activated, downstream signaling is engaged to induce the transcription factors in the nucleus, which, in turn, promote the expression of target genes, including types I and III IFNs, and a number of other important pro-inflammatory cytokines [23]. Among the transcriptional factors involved, IRF3 and NF-κB play a central role [24], with the IRF3 protein being involved in the production of interferons [25], while NF-κB is mainly employed in the induction of the proinflammatory response [26]. Even if both IRF3 and NF-κB are reported to be crucial in RNA sensing signaling, they are differentially induced by endosomal TLRs-3 and -7. In fact, while TLR7 activation leads mainly to NF-κB recruitment, TLR3 typically activates both NF-κB and the IRF3 signal [27]. This differential signaling is possible because both TLR3 and TLR7 involve the kinase TBK1, which is responsible for IRF3 and NF-κB phosphorylation. This first signal is followed by a second one that is addressed to all the surrounding cells, which are led to express a great number of interferon-stimulated genes, in order to establish the antiviral state [28].

In this study, we used Calu-3/MRC-5 multicellular spheroids as an in vitro lung model. The choice to use MTCSs lung model was supported by the evidence that both epithelial and fibroblast components could participate in the inflammatory response observed during lung damage [29], contributing to both the cytokine storm and antiviral response.

Our aim was to investigate how PRRs activation during the SARS-CoV-2 infection may affect the innate immune response in the lung cell environment.

## 2. Materials and Methods

### 2.1. Cell Cultures

African green monkey kidney epithelial Vero E6 (ATCC CRL-1586), human lung fibroblast MRC-5 cells (ATCC CCL-171) (LGC Standards S.r.l., Milan, Italy) and human lung adenocarcinoma Calu-3 cells (ATCC, HTB-55), were grown in EMEM (ThermoFisher Scientific, Milan, Italy) with 1% penicillin–streptomycin (ThermoFisher Scientific, Milan, Italy), 1% L-glutamin (ThermoFisher Scientific, Milan, Italy) and 10% fetal bovine (ThermoFisher Scientific, Milan, Italy), and cultured at 37 °C in presence of 5.0% CO_2_.

### 2.2. 3D Cultures:Multicellular Spheroids Formation

Calu-3/MRC-5 multicellular spheroids (MTCSs) were obtained using the liquid overlay method [30]. Briefly, Calu-3 and MRC-5 cells were seeded in 1:5 ratio to obtain a total of 5000 cells/well in 200 uL of complete EMEM (ThermoFisher Scientific, Milan, Italy) into a 96-well plate (Nunc, ThermoFisher Scientific, Milan, Italy) previously filled with 1.5% sterile agarose (ThermoFisher Scientific, Milan, Italy). In order to check cell distribution into MTCSs, Calu-3 and MRC-5 single-cell suspensions were stained, respectively, with Syto9 (green fluorescent nucleic acid stain) and Syto59 (red fluorescent nucleic acid stain) (ThermoFisher Scientific, Milan, Italy). After seeding, the plate was briefly centrifuged (200g × 1min) and the plate was incubated at 37 °C in presence of 5.0% CO_2_ to allow cell aggregation and spheroids formation. After 3–4 days of culture, a single spheroid was formed into each well and checked by immunofluorescence for cell distribution into spheroids. The MTCSs obtained were, on average, 250–300 um in diameter.

### 2.3. MTT Assay for Cell Viability

Cell viability of MTCSs was assessed by MTT assay (Roche Diagnostics, Milan, Italy) after SARS-CoV-2 infection as previously described [30]. Briefly, 10 uL of MTT solution was added to each well overnight. The day after, 100 uL of solvent was added and after 4 h the absorbance at 570 nm was measured. Results are expressed as mean value ± SD percent optical density (OD) derived from three independent experiments.

### 2.4. SARS-CoV-2 Propagation and Infection

SARS-CoV-2 was isolated from a nasopharyngeal swab retrieved from a patient with COVID-19 (Caucasian man of Italian origin, genome sequences available at GenBank (SARS-CoV-2-UNIBS-AP66: ERR4145453). This SARS-CoV-2 isolate clustered in the B1 clade, which includes most of the Italian sequences, together with sequences derived from other European countries and the United States. As previously described, the viral titer was determined by plaque assay in Vero E6 cells [30]. SARS-CoV-2 manipulation was performed in the BSL-3 laboratory of the University of Ferrara, following the biosafety requirements and accordingly with the Institutional Biosafety Committee. Both Calu-3 and MRC-5’s susceptibility to SARS-CoV-2 infection was assayed by infecting single-type cell with an MOI of 1 for 2 h at 37 °C, as previously reported (approx. 2 × 10^5^ infectious virus particles per well) [30]. SARS-CoV-2 infection in Calu-3/MRC-5 MTCSs was performed by transferring the spheroids into a new U-bottom 96 well, using a multiplicity of infection (MOI) of 1.0 for 1 h at 37 °C on a shaker. Then, 24, 48, 72 and 96 h after infection, the infected spheroids were collected and used for the different assays.

### 2.5. Viral RNA Detection

RNA extraction was performed 24 and 48 h post-infection (hpi) with MagMAX viral/pathogen nuclei acid isolation kit (Thermo Fisher, Milan, Italy), a kit for the recovery of RNA and DNA from virus, as previously described [14]. SARS-CoV-2 titration was obtained by TaqMan 2019nCoV assay kit v1 real-time qPCR (Thermo Fisher, Milan, Italy).

### 2.6. MTCSs Treatment for RNA Sensor Pathways Analysis

The evaluation of RLRs and TLRs inducible expression was performed using the following RLRs and TLRs agonists: RIG-I/MDA5 agonist 5′ triphosphate hairpin RNA complexed with transfection reagent LyoVec (1 µg/mL) (Invivogen, San Diego, CA, USA); TLR3 agonist Poly(I:C) (HMW) (2 µg/mL) (Invivogen, San Diego, CA, USA); TLR7/8 agonist—imidazoquinoline compound R848 (2 µg/mL) (Invivogen, San Diego, CA, USA); TLR4 agonist lipopolysaccharides LPS-B5 (LPS from E. coli 055:B5) (1 µg/mL) (Invivogen, San Diego, CA, USA). RNA sensors inhibition was performed using the following RLRs antagonists: TLR3/dsRNA complex inhibitor (30 uM) (Sigma-Aldrich, Merck Life Science S.r.l., Milan, Italy); TLR7 inhibitor Pepinh-MYD (50 uM) (Invivogen, San Diego, CA, USA). For the evaluation of the role of IRF3 and NF-κB we used MRT67307 (20 nM) (Sigma-Aldrich, Merck Life Science S.r.l., Milan, Italy), which prevents IRF3 phosphorylation, and helenalin (10 μm) (Cayman Chemicals, Ann Arbor, MI, USA), an inhibitor of NF-κB.

Further, siRNAs specific to human TLR3 (assay ID 107054; Thermo Fisher, Milan, Italy) and human TLR7 (assay ID 108895; Thermo Fisher, Milan, Italy), and the non-specific control siRNA (Ambion Silencer Negative Control) (Thermo Fisher, Milan, Italy), were transfected to Calu-3/MRC5 cells cultured on a 6-well plate (6 × 10^5^/well) using the Lipofectamine RNAiMAX reagent (Thermo Fisher, Milan, Italy) according to the manufacturer’s instructions.

### 2.7. Gene Expression Analysis

RNA sensors pathway genes expression was evaluated on RNA extracted by using the RNeasy kit (Qiagen, Milan, Italy). DNase treatment was used to check for contaminant DNA presence, using β-actin PCR as a control. RT2 first strand kit (Qiagen, Milan, Italy) was used for RNA reverse transcription and cDNAs were immediately used or stored at −20 °C. Gene expression analysis on extracted RNA was performed by real-time quantitative PCR using PowerUp SYBR Green Master Mix (Thermo Fisher, Milan, Italy) and the primer sets reported in Table 1.

Amplification followed this fast protocol, as follows: 1 cycle at 50 °C for 2 min, 1 cycle at 95 °C for 2 min and 40 cycles at 95 °C for 1 s and 60 °C for 30 s. Quantitative PCR analysis was performed using QuantStudio3 real-time PCR detection system (Applied Biosystems, Thermo Fisher, Milan, Italy). Relative quantification of given mRNA levels for the samples was conducted using the 2^−ΔΔCT^, 2 (Delta Delta CT) method [31] normalized to the constitutively expressed housekeeping gene GAPDH. Relative fold changes were generated comparing the non-infected control (NT) to the samples.

### 2.8. Immunofluorescence Analysis

Spheroids were air-dried, fixed in cool methanol at –20 °C for 10 min. After rehydration in PBS, MTCS were permeabilized with PBS—3%, BSA—0.1%, TritonX for 30 min at RT and then incubated with a specific antibody directed against SARS-CoV-2 nucleocapsid protein (NP) (MA1-7404, Thermo Fisher, Milan, Italy) or against human angiotensin-converting enzyme 2 (ACE2) (SN0754 Thermo Fisher; Italy) as previously described [32], followed by incubation with FITC goat anti-mouse IgG (H + L) secondary antibody (Thermo Fisher, Milan, Italy). Immunofluorescence was visualized by fluorescent microscopy (Nikon Eclipse Nikon Eclipse TE2000S, Milan, Italy). DNA was stained using DAPI (Thermo Fisher, Milan, Italy).

All MCTS measured 500 μm +/− 20 μm in diameter and were subdivided into seven parts (40, 80, 120, 160, 200, 240, 290 μm) on the basis of the distance from the surface of the spheroid. The number of NP-positive cells was determined in each part of the spheroid and expressed as percentage of NP-positive cells in the spheroid area.

### 2.9. Wesern Blot Analysis

TLR3 and TLR7 protein expression were quantified by Western blot assay. Whole cell lysates were treated with RIPA buffer containing proteinase inhibitor cocktail (Sigma-Aldrich, Merck Life Science S.r.l., Milan, Italy). Protein contents were evaluated by means of the Bradford assay (Bio-Rad, Milan, Italy) using bovine albumin (Sigma-Aldrich) as standard. Then, 20 µg of total proteins were loaded in each well and evaluated in denaturing conditions in 10% TGX pre-cast gel (Bio-Rad, Milan, Italy), with subsequent electroblotting transfer onto a PVDF membrane (Millipore, Merck Life Science S.r.l., Milan, Italy). The membrane was incubated with a specific antibody for the protein to be analyzed, then with a horseradish peroxidase (HRP)-conjugated anti-mouse antibody (1:5000; Amersham Biosciences, Piscataway, NJ, USA) and developed with the ECL kit (Amersham Biosciences, NJ, USA). The images were acquired by Geliance 600 (Perkin Elmer, Milan, Italy). The specific antibodies used were as follows: anti-TLR3 (clone 27N3D4), anti-TLR7 (clone NBP2-24905) (Novus Biologics, Milan, Italy). The complete Western blots are reported in Appendix A.

### 2.10. IRF3 and NF-κB Expression and Phosphorilation Analysis

The evaluation of IRF3 and NFκB expression and phosphorylation status was performed using the detection kit human total IRF-3 and phospho-IRF-3 (S386) ELISA kit (RayBiotech, Peachtree Corners, GA, USA), and total NF-κB p65 and phospho-NF-κB p65 (S536) (Abcam, Cambridge, UK) on MTSC cell lysates. MTCS were lysed for 30 min on ice in modified RIPA buffer with 150 mM NaCl, 1% Nonidet P-40, 50 mM Tris-HCl, pH 7.4, 1 mM Na3VO4, 0.25% sodium deoxycholate, and 1 mM NaF supplemented with protease inhibitor cocktails (Roche Diagnostics Corporation, Mannheim, Germany). Total protein extract was collected from supernatant after centrifugation at 12.000× *g* for 20 min at 4 °C. Protein content was evaluated by Bradford’s method, with bovine serum albumin as calibrator.

### 2.11. Soluble Factors Quantification by ELISA Assay

IL-1 α, IL-1 β, IL-4, IL-6, IL-10, interferon-α (IFN-α), interferon-β (IFN-β), interferon-γ (INF-γ), interferon-lamba1 (IL-29), interferon-λ2 (IL28A), interferon-λ3 (IL-28B) levels were evaluated in MTCSs culture supernatants by single ELISA kit assays (myBiosource, San Diego, CA, USA) following the customer’s protocols.

### 2.12. Statistical Analysis

Two-tailed Student’s *t*-test was used for comparative analysis between individual parameters, relative expression of target genes normalized to the expression of GADPH and for soluble factors evaluation, expressed as fold change relative to the corresponding control group. The data were analyzed by paired Student’s *t*-test. *P* values < 0.05 were considered significant. The statistical analyses were performed with GraphPad Prism version 9 software (GraphPad, La Jolla, CA, USA).

## 3. Results

### 3.1. Calu-3/MRC-5 Multicellular Spheroids Are Efficiently Infected by SARS-CoV-2

The main structural cell types of the lung, epithelial and fibroblast cells were cultured in a 3D in vitro model, which was obtained 4 days after seeding the Calu3 and MRC-5 cells in a ratio of 1:5 (Figure 1a). In order to check the cell distribution in the multicellular spheroid (MTCS), Calu-3 and MRC-5 single-cell suspensions were pre-stained with Syto9 and Syto59, respectively. We observed the localization of the MRC-5 cells in the core of the spheroid, while the Calu-3 cells were in the outer region. This peculiar cellular distribution is in line with the results reported in the literature [32], and is consistent with the structure of the airway epithelium. We observed a high viability rate until 72 h of culturing (Figure 1b), while at 96 h of culturing, the spheroids presented a reduced viability (Figure 1b; *p* < 0.001; Student’s *t*-test). We used these MTCSs as an in vitro model for lung SARS-CoV-2 infection.

MTCSs were infected with 1.0 MOI of SARS-CoV-2 for 2 h and viral titration was performed by real-time qPCR. We observed susceptibility and permissivity to the SARS-CoV-2 infection, with a significant increase in the viral load 48 h after the infection (Figure 1c, d; *p* < 0.0001; Student’s *t*-test). At 96 h post-infection, we had a significant decrease in the viral titration (Figure 1d), due to the senescence of the spheroids, as previously observed (Figure 1b). Figure 1c shows the expression of the SARS-CoV-2 NP protein in MTCS. We estimated the percentage of SARS-CoV-2 NP-positive cells on the basis of the distance to the spheroid surface. The MCTS measured 500 μm +/− 20 μm in diameter, and we subdivided the spheroid into seven parts, on the basis of the distance from the surface of the spheroid. We observed a progressive decrease in the proportion of NP-positive cells from the surface (40, 80, 120 μm) to the center of the spheroid (160, 200, 240, 290 μm) (Figure 1c,e). Approximately 60% of cells were positive for NP in the outer 40 μm of the spheroid, and 50% in the outer 80 μm of the spheroid, 48 h post-infection (Figure 1c,e). Conversely, we observed that only 20% of the cells were infected in the outer 200 μm of the spheroid, and there was almost the same absence of NP positivity in the center of the spheroid. The observed regionalization of SARS-CoV-2 infection in the spheroids might depend on the cell position to the spheroid surface and/or their differential permissivity to the virus.

### 3.2. Calu-3 and MRC-5 Permissivity to SARS-CoV-2 Infection

To evaluate the possible different permissivity of these two cell types, the Calu-3 and MRC-5 cells were infected with 1.0 MOI of SARS-CoV-2 inoculum. We assayed the viral infection in the culture supernatants 48 h post-infection (pi), by real-time qPCR. We observed that both the cell lines are susceptible and permissive to the SARS-CoV-2 infection (Figure 1f), as already reported [33]. The MRC-5 cell line presented a lower permissivity in comparison with the Calu-3 cell line, as demonstrated by the difference of more than 1log viral load in the infected cell supernatants (Figure 1f; *p* < 0.001; Student’s *t*-test). To account for this difference, we evaluated the expression of the SARS-CoV-2 receptor human ACE2 (hACE2) on both cell lines. As reported in Figure 1g,h, both the cell lines express hACE2 on the cell surface. The relative intensity measurement of immunofluorescence showed that Calu-3 expressed higher levels of hACE2 in comparison with the MRC-5 cell line (Figure 1h). This difference might influence the different SARS-CoV-2 permissivity of these two cell lines, together with their positions in the MTCS, which resemble the in vivo condition.

### 3.3. Calu-3/MRC-5 MTCS Response to SARS-CoV-2 Infection

One of the main cellular response systems to the coronavirus infection might be the activation of the RNA sensor pathways. We selected to evaluate the most important RNA sensors (TLR3, TLR7, TLR8, RIG-I, MDA5) in MTCS infected with SARS-CoV-2. TLR4 was used as a control, as implicated in bacterial lipopolysaccharide sensing. To be sure that all these pathways are expressed in the in vitro system, we treated the MTCS with RNA sensors agonists. We obtained the activation of all the evaluated RNA sensors (Appendix A). After 48 h of infection, we observed a predominant induction of both TLR7 and TLR3 expression (Figure 2a,b; *p* < 0.001; Student’s *t*-test). On the contrary, the TLR4, TLR8 and RLRs genes (RIG-I, MDA5) expression was not significantly modified by the SARS-CoV-2 infection (Figure 2a). The TLR3 and TLR7 protein expression was similarly increased in the presence of SARS-CoV-2 infection (Figure 2b,c; *p* < 0.01; Student’s *t*-test). These data support an induction of both TLR7 and TLR3 RNA sensing during SARS-CoV-2 infection.

### 3.4. SARS-CoV-2 Infection Induced an Increase in Cytokines and Interferon Secretion

One of the critical points in SARS-CoV-2 infection is the establishment of a strong pro-inflammatory environment, the so-called cytokine storm. The cytokine storm might also be induced by the RNA sensing activation. We selected the most important cytokines involved in the COVD19-associated cytokine storm, including IL-1 α, IL-1β, IL-4, IL-6, IL-10, and interferons, in order to evaluate the effect of SARS-CoV-2 infection on their induction in MTCS [34]. We observed an increase in IL-1 α, IL-1β, IL-4, and IL-6 levels in SARS-CoV-2 48 h-infected cells, in comparison with uninfected MTCS (Figure 3a–d) (*p* < 0.001; Student’s *t*-test). The levels of these inflammatory cytokines are reduced in the presence of the TLR3 inhibitor (Figure 3a–d) (*p* < 0.001; Student’s *t*-test). On the contrary, the IL-10 levels were not affected by the SARS-CoV-2 infection (Figure 3e). Then, we evaluated the expression of type I and type II interferons. IFN- α and INF-β were induced 24 h post-infection and also maintained a high secretion 48 h post-infection (Figure 3f,g) (*p* < 0.001; Student’s *t*-test). Meanwhile, the addition of the TLR3 inhibitor reduced the secretion of IFN-α and IFN-β 24 h post-infection, while the addition of the TLR7 inhibitor reduced the expression of these type I IFNs 48 h post-infection (Figure 3f,g) (*p* < 0.001; Student’s *t*-test). IFN-γ was induced by SARS-CoV-2 48 h post-infection (Figure 3h) (*p* < 0.001; Student’s *t*-test) and reduced by the TLR7 inhibitor (Figure 3h) (*p* < 0.001; Student’s *t*-test). IFN-λ1 and IFN-λ2 were not modified by SARS-CoV-2 infection (Figure 3i,j), while IFN-λ3 was induced after 48 h of SARS-CoV-2 infection (Figure 3k) (*p* < 0.001; Student’s *t*-test) and reduced by the TLR7 inhibitor (Figure 3k) (*p* < 0.001; Student’s *t*-test).

These results suggest that TLR3 is mainly implicated in cytokine secretion control and type 1 IFN expression 24 h post-infection; TLR7 controls the expression of type 1 IFN, IFN-γ, and IFN-λ3 expression in the late phases of SARS-CoV-2 infection.

### 3.5. TLR3 and TLR7 Activation Followed Different Signal Pathways after SARS-CoV-2 Infection

To evaluate the proposed effect of SARS-CoV-2 on TLR3- and TLR7-mediated gene expression, we assessed the expression of the following TLR3- and TLR7-associated key transcription factors: NF-κB, which induces TLR-dependent gene activation, and IRF3, which mediates TLR3-dependent gene expression [35]. In SARS-CoV-2-infected cells, there was a significant increase in IRF3 expression 24 h post-infection, and of NF-κB 48 h post-infection, which was maintained until 72 h post-infection (Figure 4a). 

To evaluate the effective activation of IRF3 and NF-κB, we assessed IRF3 Ser386 phosphorylation, which induces dimerization and association with the coactivators CREB-binding protein/p300, and the NF-κB p65 Ser536 phosphorylation that leads to the nuclear localization of the transcriptionally active complex. Both NF-κB and IRF3 presented an increased phosphorylation 48 h post-infection (Figure 4b,e; *p* < 0.001; Student’s *t*-test). To assess the specificity of NF-κB and IRF3 induction by TLR3 and TLR7, we treated MTCs with TLR3 and TLR7 inhibitors, or RNA silencing. The TLR3/dsRNA complex inhibitor and TLR7 inhibitor Pepinh-MYD did not affect TLRs protein expression (Figure 2b,c), but blocked their activation and, consequently, NF-κB and/or IRF3 phosphorylation. In particular, NF-κB phosphorylation was reduced after TLR3 and TLR7 inhibition (Figure 4c) (*p* < 0.001; Student’s *t*-test), while IRF3 phosphorylation was decreased only after TLR3 inhibition (Figure 4e) (*p* < 0.001; Student’s *t*-test). TLR3- and TLR7-specific siRNA transfection resulted in the absence of RNA (Appendix A) and protein expression (Figure 2b,c). Similarly, NF-κB phosphorylation was reduced after TLR3 and TLR7 silencing (Figure 4c) (*p* < 0.001; Student’s *t*-test), while IRF3 phosphorylation was decreased only after TLR3 silencing (Figure 4e) (*p* < 0.001; Student’s *t*-test).

Summarizing these results, TLR3 might act via NF-κB and IRF3, while TLR7 mainly acts via NF-κB activation. To investigate the role of NF-κB and IRF3 activation in TLR3- and TLR7-mediated inflammatory cytokine and interferons gene expression, during SARS-CoV-2 infection, we used MRT67307, which prevents IRF3 phosphorylation and expression of interferon-stimulated genes, and helenalin, an inhibitor of NF-κB (Figure 5). The pretreatment of MTCS with MRT67307 reduced IFN-α and IFN-β gene expression (Figure 5e; *p* < 0.001; Student’s *t*-test), and slightly reduced IL-6 and IL-4 (Figure 5c, d; *p* < 0.012, *p* = 0.023, respectively; Student’s *t*-test), while NF-κB inhibition, by helenalin, completely abrogated the IL-1 α, IL-1β, IL-4, and IL-6 gene expression (Figure 5a–d; *p* < 0.001; Student’s *t*-test), and reduced type 1 IFN, IFN-γ and IFN-λ gene expression (Figure 5f; *p* < 0.0001; Student’s *t*-test).

## 4. Discussion

The role of the RNA sensor pathways during SARS-CoV-2 infection is of extreme interest, as TLRs are the innate mediators of the anti-viral response and might influence the pathogenesis of SARS-CoV-2 infection. TLR3 pathway activation is associated with the production of IFN-β by macrophages in murine coronavirus infection [36]; TLR7 and TLR8 activation enhances a cytokine storm in SARS-CoV-1 infection, causing several side effects [37].

Our results showed that the SARS-CoV-2 infection of Calu-3/MRC-5 multicellular spheroids induces the activation of TLR3 and TLR7 RNA sensor pathways. In particular, TLR3 might act via IRF3-producing IFN-α and IFN-β during the first 24 h post-infection. Then, TLR3 activates the NFκB transduction pathway, leading to pro-inflammatory cytokine secretion (IL-1α, IL-1β, IL-4, IL-6). Conversely, TLR7 seems to act mainly via NFκB, inducing type 1 IFN, IFN-γ, and IFN-λ3, starting from the 48 h post-infection.

These data suggest a differential timing of TLRs activation, which, on one hand, might interfere with SARS-CoV-2 infection, activating the host immune response, or, on the other hand, might lead to a cytokine storm, with an adverse effect on disease follow-up. Totura et al. observed that the induction of TRIF-driven and MyD88-driven pathways by TLRs are essential in the control of SARS-CoV infection [38]. As a proof of concept, TLR3/TLR4 double-negative mice were more susceptible to SARS-CoV infection, and the deletion of TRIF increased the SARS-CoV-dependent risk of mortality. Further, van der Made et al. showed a TLR7 loss-of-function variant in four male patients with severe COVID-19 infection that presented an impaired type I and type II IFNs response. Interestingly, SARS-CoV-2 has more ssRNA motifs that could be recognized by TLR7 [39], inducing a strong pro-inflammatory response [40,41].

These data support the harmful and beneficial role of TLRs in SARS-CoV-2 infection. Our data clarify the role of the TLR components, supporting the potential use of TLRS antagonists and agonists as therapeutic tools in SARS-CoV-2 infection. Not only are TLRs important, but also the related pathways. A study on SARS-CoV, which was responsible for the worldwide outbreak of SARS in 2003, showed that the SARS-CoV nucleocapsid protein (N protein) activates NF-κB in Vero E6 cells, in a dose-dependent manner [42]. DeDiego et al. proved that inhibitors of the NFκB pathway increased the survival rate in both in vitro and in vivo studies, using mice with reduced lung pathology [43]. In vitro studies in the previous SARS epidemic have shown that the spike (S) protein induces a strong cytokine response in infected mononuclear cells, through the NFκB pathway. SARS-CoV-2 is more sensitive to interferon treatment [44], less efficient in suppressing cytokine induction via IRF3 nuclear translocation [45], and permissive of a higher level of induction of interferon-stimulated genes, in comparison with SARS-CoV [46]. Our data support an implication of both the transcription factors, which have an important role in controlling cytokine and IFN expression during SARS-CoV-2 infection. In conclusion, our data suggest an important role for TLR3 and TLR7 in COVID-19 disease, with a definition of the possible transduction pathways and activation timing. The suppression of excessive activation of TLRs seems to have a role in SARS-CoV-2 infection, as supported by several clinical trials [37], with the purpose of controlling TLRS activation and, consequently, SARS-CoV-2 infection.

This was confirmed by the recent findings on the contribution of both TLR3 and TLR7 in the antiviral signal against SARS-CoV-2 and that, in presence of genetic loss-of-function variants of TLR7, the TLR3 signal was not affected [22].

## Figures and Tables

**Figure 1 microorganisms-09-01820-f001:**
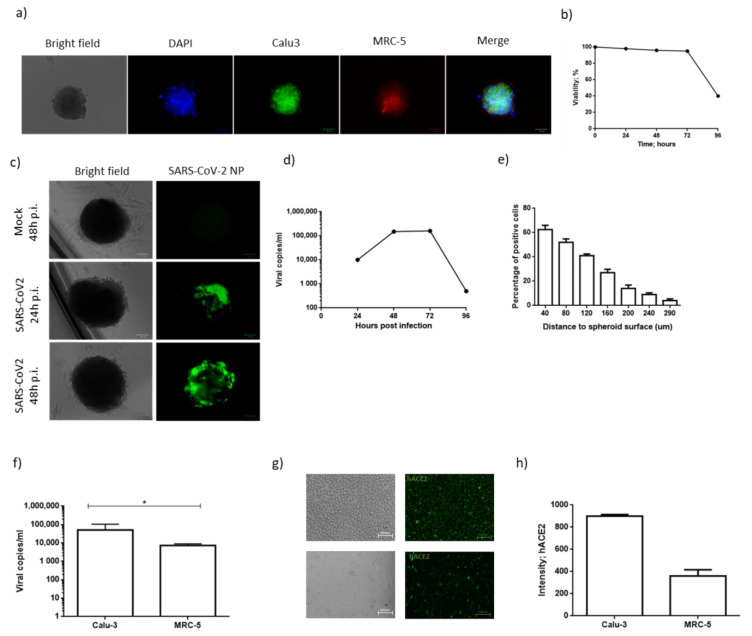
(**a**) A representative Calu-3/MRC-5 multicellular spheroid, 48 h after culture. Calu-3 and MRC-5 single-cell suspensions were pre-stained with Syto9 and Syto59, respectively. DAPI was used for nuclear staining. Scale bar 100 µm. (**b**) Graphic representation of the percentage of viable cells during a time lapse of 96 h. Cell viability was measured by MTT assay. (**c**) Representative Calu-3/MRC-5 multicellular spheroids stained with anti-SARS-CoV-2 nucleoprotein. UV-inactivated SARS-CoV-2 (mock) was used as negative control. Scale bar 100 µm. (**d**) Calu-3/MRC-5 multicellular spheroids were infected with SARS-CoV-2 at a multiplicity of infection (MOI) of 1.0 for 1 h at 37 °C. Thereafter, the cells were washed and cultured for 24, 48, 72 or 96 h. Viral yield was quantified in the cell supernatant using quantitative reverse-transcription PCR (RT-qPCR). At least three independent replicates were tested. Data represent three independent experiments. (**e**) Percentage of NP-positive cells, subdivided according to the distance to spheroid surface. Data are expressed as mean +/− standard deviation. (**f**) Calu-3 or MRC-5 cells were SARS-CoV-2 at a multiplicity of infection (MOI) of 1.0. Viral yield was quantified in the cell supernatant using quantitative reverse-transcription PCR (RT-qPCR). At least three independent replicates were tested. Data are representative of three independent experiments. (**g**) hACE2 staining of Calu-3 (upper panel) and MRC-5 (lower panel) cells. Scale bar 100 µm. (**h**) Levels of hACE2 staining in Calu-3 and MRC-5 cells. Data correspond to the mean +/− standard deviation. * *p* value < 0.05, calculated with Student’s *t*-test.

**Figure 2 microorganisms-09-01820-f002:**
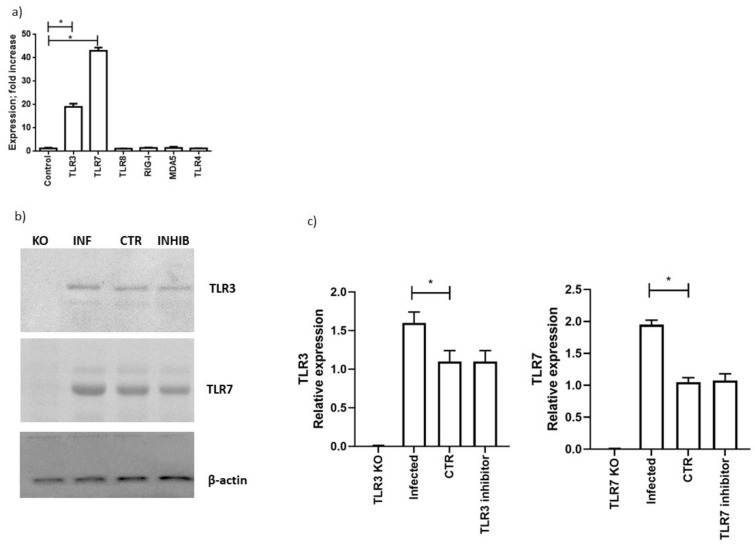
(**a**) Calu-3/MRC-5 multicellular spheroids were infected with SARS-CoV-2 at a multiplicity of infection (MOI) of 1.0 for 1 h at 37 °C. Thereafter, the cells were washed and cultured for 48 h. Levels of expression, quantified as fold increase in comparison with uninfected cells of TLR3, TLR7, TLR8, RIG-1, MAD5 and TLR4 are reported and are representative of three independent experiments. (**b**) Western blot analysis of TLR3, TLR7 and β-actin protein expression in Calu-3/MRC-5 multicellular spheroids (CTR), silenced for TLR3 (TLR3 KO), TLR7 (TLR7 KO) with RNA silencing technology; infected with SARS-CoV-2 with at a multiplicity of infection (MOI) of 1.0 (INF: infected) and treated with TLR3 or TLR7 inhibitors (INHIB: inhibitor). The molecular weights were determined by protein ladder (Bio-Rad, Milan, Italy). Actin was evidenced at 44 kDa, TLR3 and TLR7 at 116 kDa. The images were acquired by Geliance 600 (Perkin Elmer, Milan, Italy). The complete Western blots are reported in Appendix A. (**c**) Evaluation of protein expression by densitometry (GelDoc software; Biorad, Italy), normalized on β-actin content. Data are representative of three independent experiments. Data correspond to the mean +/− standard deviation. * *p* value < 0.05, calculated with Student’s *t*-test.

**Figure 3 microorganisms-09-01820-f003:**
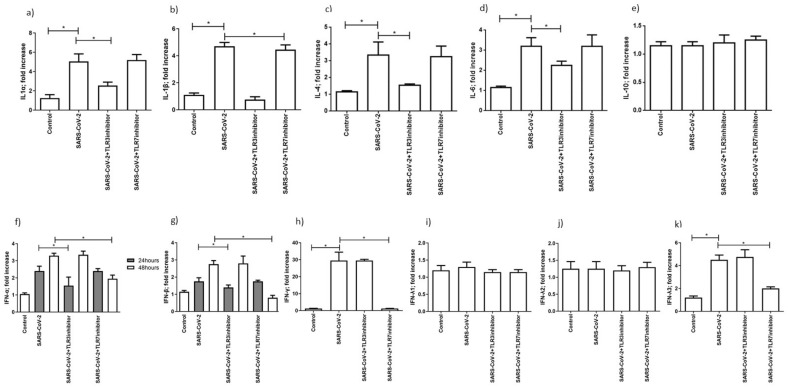
Levels of expression of cytokines (**a**) IL-1α, (**b**) IL-1β, (**c**) IL-4, (**d**) IL-6 and (**e**) IL-10 are reported after 48 h post-infection. The levels after 24 h post-infection were under the detection limit of the assays. Levels of (**f**) IFN-α, (**g**) IFN-β are reported 24 and 48 h post-infection. Data correspond to the mean +/− standard deviation. * *p* value < 0.05, calculated with Student’s *t*-test. Levels of (**h**) IFN-γ, (**i**) IFN-λ1, (**j**) IFN-λ2, (**k**) IFN-λ3 are reported 48 h post-infection. The levels after 24 h post-infection were under the detection limit of the assays. Data correspond to the mean +/− standard deviation. * *p* value < 0.05, calculated with Student’s *t*-test.

**Figure 4 microorganisms-09-01820-f004:**
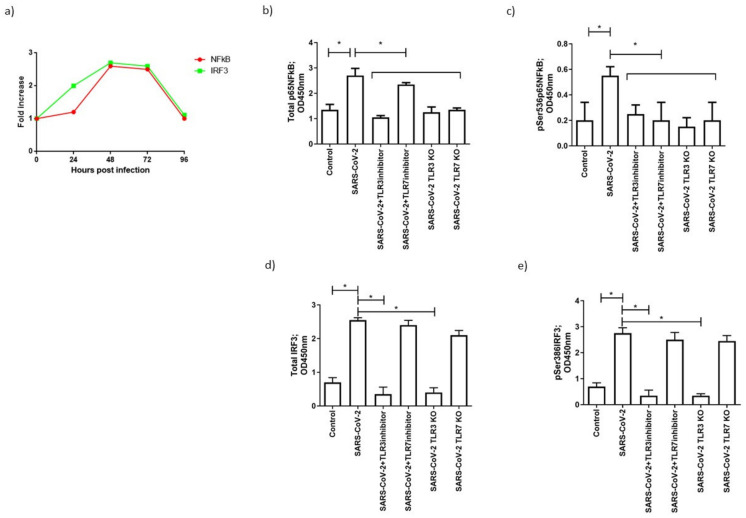
(**a**) Levels of expression, quantified as fold increase in comparison to uninfected cells, of NF-κB and IRF3, are reported and are representative of three independent experiments. Levels of (**b**) total and (**c**) phosphorylated NF-κB and of (**d**) total and (**e**) phosphorylated IRF3 in Calu-3/MRC-5 multicellular spheroids infected with SARS-CoV-2 and analyzed 48 h post-infection, with or without inhibitor (inhibitor) or silencing (KO) treatment. Data correspond to the mean +/− standard deviation. * *p* value < 0.05, calculated with Student’s *t*-test.

**Figure 5 microorganisms-09-01820-f005:**
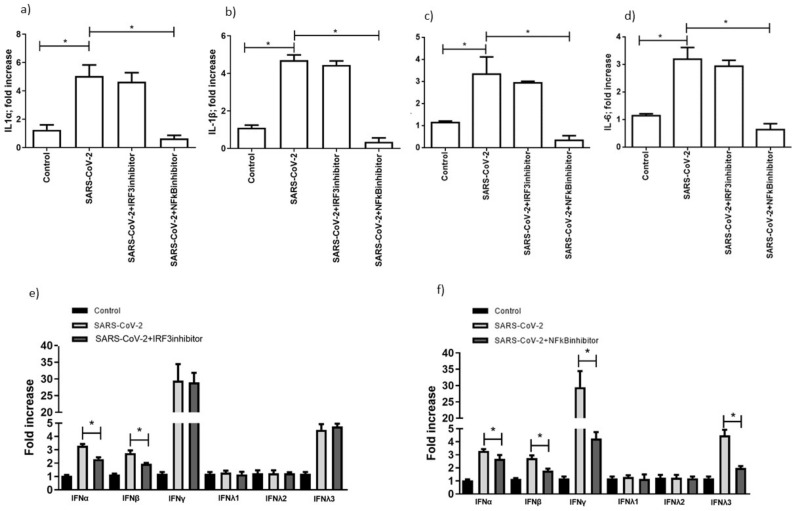
Levels of expression of cytokines (**a**) IL-1α, (**b**) IL-1β, (**c**) IL-4, (**d**) IL-6, (**e**), (**f**) IFN-α, IFN-β, IFN-γ, IFN-λ1, IFN-λ2, IFN-λ3 are reported, in the presence or absence of NFκB or IRF3 inhibitors. The levels of IFNs in the presence of IRF3 inhibitor were evaluated 24 h post-infection. Data correspond to the mean +/− standard deviation. * *p* value < 0.05, calculated with Student’s *t*-test.

**Table 1 microorganisms-09-01820-t001:** PrimeTime qPCR assays used for gene expression analysis.

Target Gene	PrimeTime qPCR Primer Assay *
RIG-I	Hs.PT.58.4273674
MDA5	Hs.PT.58.1224165
TLR3	Hs.PT.58.25887499.g
TLR4	Hs.PT.58.38700156.g
TLR7	Hs.PT.58.39183219.g
TLR8	Hs.PT.58.15023918.g
IRF3	Hs.PT.58.27933933.g
NF-κB	Hs.PT.58.20344216
GAPDH	Hs.PT.58.25887499.g

* PrimeTime qPCR primer assays provide a primer pair designed for real-time PCR using intercalating dyes, such as SYBR® Green (IDT, Leuven, Belgium).

## Data Availability

The data presented in this study are all available upon request.

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
