# Peer review of "TLR3 and TLR7 RNA Sensor Activation during SARS-CoV-2 Infection"

_microorganisms, 2021, doi:10.3390/microorganisms9091820_

Round 1
Reviewer 1 Report
The Authors investigate the role of RNA sensing receptors in recognition of SARS-CoV-2 infection and activation of the innate immune response in lung cells.
Minor points:
- Nomenclature should be standardized in the text (e.g. NF-κB, SARS-CoV-2)
- Line 65. Names of particular TLRs receptors involved in sensing of viruses RNA should be added, as it was done for RLRs receptors.
- Lines 67-76. There lacks some introduction to the role of TLR3 in the context of virus infection as it was described for TLR8 and TLR7
- Line 95-96. The sentence "Our aim was to investigate how PPRs activation during SARS-CoV-2 infection may affect the PPRs activation in lung cell environment" is not clear and must be reformulated.
- Line 100. Human lung adenocarcinoma
- Line 139. I assume that it should be “The evaluation of RLRs and TLRs inducible expression was performed using RLRs and TLRs agonists” instead of existing sentence.
- Line 142-143. LPS is TLR4 agonist
- Line 209. Incorrect citation
- Figure 3, 4 and 5 description should be corrected.
- Figure 4. On the graph 4a and 4b NF-κB p65 should be put instead of NF-κB
Major points:
- Line 218-220. It is not clear for me how this estimation was performed. It lacks some additional information about the procedure in Materials and Methods section.
- Figure 4. Graphs from Supplementary Figure S2 are more informative than graphs from figure 4a and 4b so the Authors should consider replacing the graphs, as the graphs in the main text do not support the conclusion that NF-κB activation is reduced after TLR3 inhibition.
- Line 367-369. According to presented results IL-1α, IL-1β, IL-6 and IL-4 expression is connected with TLR3 (Figure 3) and NF-κB (Figure 5) activation. Expression levels of this cytokines is presented after 48 hours post-infection as the levels after 24 hours post-infection was under the detection limit of the assay. So why the Authors conclude that “In particular, TLR3 might act via IRF3 producing IL-1alpha, IL-1beta, IL-4, IL-6 during the first 24 hours post infection” ?
These changes will improve the manuscript
Author Response
We thank the reviewer for the valuable requests for modification. We believe that their comments have increased the value of the manuscript. The modifications have been highlighted in the manuscript and the point-to-point responses are reported below.
Sincerely.
Roberta Rizzo
Reviewer #1
Minor points:
- Nomenclature should be standardized in the text (e.g. NF-κB, SARS-CoV-2)
A: We have carefully revised the manuscript.
- Line 65. Names of particular TLRs receptors involved in sensing of viruses RNA should be added, as it was done for RLRs receptors.
A: We have added the TLRs implicated in anti-viral sensing. (Page 2; Line 66)
- Lines 67-76. There lacks some introduction to the role of TLR3 in the context of virus infection as it was described for TLR8 and TLR7
A: We agree with the reviewer. We have added a description also for TLR3 in the context of virus infection. (Page 2; Line 69)
- Line 95-96. The sentence "Our aim was to investigate how PPRs activation during SARS-CoV-2 infection may affect the PPRs activation in lung cell environment" is not clear and must be reformulated.
A: We have revised the sentence.
- Line 100. Human lung adenocarcinoma
A: Corrected
- Line 139. I assume that it should be “The evaluation of RLRs and TLRs inducible expression was performed using RLRs and TLRs agonists” instead of existing sentence.
A: We thank the reviewer for the suggestion.
- Line 142-143. LPS is TLR4 agonist
A: Sorry for the typos. We have corrected the mistake.
- Line 209. Incorrect citation
A: Sorry for the typos. We have corrected the mistake.
- Figure 3, 4 and 5 description should be corrected.
A: The Figures legends have been revised.
- Figure 4. On the graph 4a and 4b NF-κB p65 should be put instead of NF-κB
A: As requested we have modified the Figure 4.
Major points:
- Line 218-220. It is not clear for me how this estimation was performed. It lacks some additional information about the procedure in Materials and Methods section.
A: We agree with the reviewer. We have clarified the procedure to estimate the infection of spheroids. (Page 5 line 187; Page 6 line 246).
- Figure 4. Graphs from Supplementary Figure S2 are more informative than graphs from figure 4a and 4b so the Authors should consider replacing the graphs, as the graphs in the main text do not support the conclusion that NF-κB activation is reduced after TLR3 inhibition.
A: As requested, we have modified the Figure 4.
- Line 367-369. According to presented results IL-1α, IL-1β, IL-6 and IL-4 expression is connected with TLR3 (Figure 3) and NF-κB (Figure 5) activation. Expression levels of this cytokines is presented after 48 hours post-infection as the levels after 24 hours post-infection was under the detection limit of the assay. So why the Authors conclude that “In particular, TLR3 might act via IRF3 producing IL-1alpha, IL-1beta, IL-4, IL-6 during the first 24 hours post infection” ?
A: The reviewer is correct. The sentence was misleading. We have corrected it. (Page 12 line 400).
These changes will improve the manuscript
A: We agree with the reviewer. The modifications have improved the quality of the manuscript.
Reviewer 2 Report
Major issues:
Do the spheroid cell lines express the RNA-sensing PRRs at protein level? The authors should know the expression before incubating them with specific agonists. Does the infection affect their expression at protein level?
There is a concern that the inhibitors used also exert non-specific effects. The specificity of IRF3 and NFkB activation by TLRs should be examined by using TLR KO cells.
Minor issues:
Ref 7 should be that of a seminal study, and not of a review.
Why production of TNF- was not investigated?
Line 65: …consist of…… (not consist in)…
Line 101: P/S?
Line 109: Explain Cyto9 and Cyto59.
Line 129: LARP?
Line 134: MagMAX Viral/Pathigen?
Line 157: Prime Time qPCR pr Assay?
Line 209: Ref 323?
Line 226: …..to contrast viral diffusion…..?
Line 247: differential; not different
Line 260: Calu-3/MRC-5 MTCS control of SARS-CoV-2 infection?
Line 318: associated
Author Response
We thank the reviewer for the valuable requests for modification. We believe that their comments have increased the value of the manuscript. The modifications have been highlighted in the manuscript and the point-to-point responses are reported below.
Sincerely.
Roberta Rizzo
Reviewer #2
Major issues:
Do the spheroid cell lines express the RNA-sensing PRRs at protein level? The authors should know the expression before incubating them with specific agonists. Does the infection affect their expression at protein level?
A: We thank the reviewer for this request. We agree on the importance to determine the protein expression of PRRs Since we observed a significant increase after SARS-CoV.2 infection only in TLR3 and TLR7, we performed the protein evaluation on these receptors. We evaluated TLR3 and TLR7 protein expression in spheroid cells with and without SARS-CoV-2infection. (Page 8 line 296; Figure 2b, c).
There is a concern that the inhibitors used also exert non-specific effects. The specificity of IRF3 and NFkB activation by TLRs should be examined by using TLR KO cells.
A: We thank the reviewer for this suggestion. We have silenced TLR3 and TLR7 receptors by RNA silencing technology, obtaining a total knock-down of protein expression. We have evaluated IRF3 and NF-kB expression and phosphorylation in these conditions, supporting the data observed with inhibitors. The results have been added to Figure 4. (Page 10 line 363).
Minor issues:
Ref 7 should be that of a seminal study, and not of a review.
A: We have changed the reference.
Why production of TNF-alpha was not investigated?
A: We evaluated also TNF-alpha levels, but they were below the detection limits of the assay. These results are in agreement with the literature (Katleen Martens, Peter W. Hellings, Brecht Steelant. Calu-3 epithelial cells exhibit different immune and epithelial barrier responses from freshly isolated primary nasal epithelial cells in vitro. Clin Transl Allergy. 2018; 8: 40), where no induction of TNF-alpha was observed in Calu-3 cells. For this, we decided not to include TNF-alpha in the cytokine panel, as it gave no informative results in this in vitro setting.
Line 65: …consist of…… (not consist in)…
A: Corrected
Line 101: P/S?
A: Penicillin-Streptomycin. We have added it to the text.
Line 109: Explain Cyto9 and Cyto59.
A: We have added the explanation “Syto9 (Green Fluorescent Nucleic Acid Stain) and Syto59 (Red Fluorescent Nucleic Acid Stain).”
Line 129: LARP?
A: LARP is the laboratory name for BSL3. We have deleted this name, as it is confusing.
Line 134: MagMAX Viral/Pathigen?
A: It is a kit is designed to recover RNA and DNA from viruses. We have added this explanation in the text.
Line 157: Prime Time qPCR pr Assay?
A: PrimeTime qPCR Primer Assays provide a primer pair designed for real-time PCR using intercalating dyes, such as SYBR® Green. We have added this information to the text.
Line 209: Ref 323?
A: It was a typo, that we have corrected.
Line 226: …..to contrast viral diffusion…..?
A: We agree with the reviewer that the sentence is misleading. We have changed the sentence into: “The observed regionalization of SARS-CoV-2 infection in the spheroids might depend on cell position to the spheroid surface and/or their differential permissivity to the virus.”.
Line 247: differential; not different
A: Corrected
Line 260: Calu-3/MRC-5 MTCS control of SARS-CoV-2 infection?
A: We have changed the heading into “Calu-3/MRC-5 MTCS response to SARS-CoV-2 infection”.
Line 318: associated
A: Corrected.
Round 2
Reviewer 1 Report
Dear Authors,
The revisions made to the manuscript and responses to suggestions and comments are appreciated and appropriate. Thank you. Please see minor comment only below:
Figures 3 and 5 legends should be corrected, e.g. in figure 3, IL-10 expression is shown on graph "e" not "f"
Author Response
We thank the reviewer for the comments.
Figures 3 and 5 legends should be corrected, e.g. in figure 3, IL-10 expression is shown on graph "e" not "f".
A: We have revised the Figures 3 and 5 legends.